# The Effect of Randomly Providing Nutri-Score Information on Actual Purchases in Colombia

**DOI:** 10.3390/nu11030491

**Published:** 2019-02-26

**Authors:** Claudio A. Mora-García, Luisa Fernanda Tobar, Jeremy C. Young

**Affiliations:** 1INCAE Business School, Campus Walter Kissling Gam, 2 km West of Vivero Proceso No. 1, La Garita, Alajuela 960-4050, Costa Rica; 2Department of Nutrition and Biochemistry, School of Sciences, Pontificia Universidad Javeriana, Carrera 7 #43-92, Office 504, Bogotá, D.C. 110231, Colombia; ltobar@javeriana.edu.co; 3Department of Business Administration, School of Economics and Management, Pontificia Universidad Javeriana, Carrera 7 #40B-36, 4th Floor, Bogotá, D.C. 110231, Colombia; jeremyyoung@javeriana.edu.co

**Keywords:** randomized field intervention, food labeling, Nutri-Score, store sales

## Abstract

Recently, front-of-package (FOP) food labeling systems have captured the attention of researchers and policy makers. Several Latin American governments are currently considering employing different FOP labeling systems. However, there is much need for more research-based evidence in these countries. In this paper, we study whether food-purchasing decisions and the nutritional qualities of those purchases are influenced by randomly informing some customers and not others about an FOP label known as Nutri-Score. We also separate the information effect from the effect of being aware of the system. We combined a randomized field intervention in a university cafeteria in Bogotá, Colombia with data from an after-purchase survey and receipts. We found that randomly providing information on Nutri-Score increased total expenditure by $0.18. Additional spending on healthier items was 21% or $0.26 higher, with no change for less healthy items. Expenditure estimates were higher among customers who were aware of the system’s existence. Customers in the study were also 10% more likely to buy a healthier item than control customers were, and the concentration of protein in their purchases was greater. Information on the Nutri-Score system increased the store’s sales. This potential financial incentive may facilitate the implementation of Nutri-Score.

## 1. Introduction

Although work on the issues of healthy food consumption and nutrition labeling started later in Latin America than in Europe [1] and the USA [2], front-of-package (FOP) nutrition label related policies in the region have developed quickly in the last 5 years [3]. The following countries have implemented a FOP system as government policy: Ecuador in 2014 (multiple Traffic Light System (TLS)), Bolivia in 2016 (multiple TLS), Mexico in 2015 (Guideline Daily Amount (GDA)) and Chile in 2016 (warning labels). Although some other Latin American governments have passed laws for FOP labeling systems, none of them has been implemented yet, namely in Argentina (Bill S-3375/15, 2015), Peru (Law 30021, 2013) and Uruguay [4]. However, there is a lack of evidence of the effectiveness of these systems in Latin American countries, including, in our context, Colombia.

Therefore, in our study, we used a randomized controlled intervention design, observing whether providing Nutri-Score label information and awareness of this system led to different purchasing decisions in a real-world environment. Our study seeks to provide evidence on FOP nutrition labeling in order to inform policy and legislation that are currently being formulated by the Colombian government to combat overweightness and obesity. 

As reported by the Pan American Health Organization (2016), the rate of overweightness and obesity in Latin America is alarming, as it is the highest in the world. In 2012, the incidence of overweightness and obesity was 62% in adults aged 20 and above. This phenomenon has become a major cause of death and disability in Latin America (55% of all causes in 2012, according to WHO Global Health Estimates [5]). 

The Colombian case is no exception, and the incidence of these conditions has significantly grown. In 2008, over 50% of adults were at least overweight, and of these, 18% were obese [6]. Awareness of this alarming issue has been rising, and in 2009, the Colombian Ministry of Social Protection (*Ministerio de Protección Social*) declared that obesity and its associated NCDs were a public health priority [7].

One effective method of combating overweightness, obesity, and NCDs is good nutrition and diet quality [8,9], even in Latin American cases [10,11]. The link between the food environment and diet quality has been observed in many studies [12,13,14,15]. So, as Surkan et al. (2016) [16] have highlighted, stores and supermarkets play an important role in affecting eating behaviors from the individual to macro levels. Nutrition labeling is one such measure that stores and supermarkets can take.

Recently, nutrition labeling has become a useful tool for influencing consumers’ food-purchasing behavior (for reviews see: [17,18,19]). Despite this, and as noted in these reviews, there is a desire for more evidence and understanding on how nutrition labeling affects real-world consumer purchases and store sales. Many of these nutrition-labeling studies use an experimental situation and measure the effect on intended consumer purchases. However, actual purchase behavior may be different from intended purchase behavior [20]. Moreover, in experimental situations, participants may act how they are expected to act, i.e.; in a socially acceptable manner (social desirability bias [21]), and not how they would actually act in real-life situations. In addition, online studies and lab experiments are highly controlled and test hypothetical treatments, which may not translate to a real-world setting. In the real world, there are factors such as competitive pricing, lack of time for decision making, and in-store marketing that are hard for experimental settings to replicate [1,22,23,24].

There are a few exceptions of studies in real-life situations measuring the effect of nutrition labels on actual food purchases (these [2,25,26,27,28,29,30,31,32,33,34,35,36,37,38]). Unfortunately, these studies normally lack a control group, and therefore, cannot support strong causal claims. Three exceptions that use controls are [16,28,32]. Nikolova & Inman (2015) [28] used a control group comparison as a robustness check in their study using scanner data from a large grocery chain before and after NuVal label implementation. Surkan, et al. (2016) [16], in a low-income neighborhood in the US, studied a multifaceted campaign that combined health education strategies with structural changes such as pricing and stocking of healthier foods, as well as promotional product labeling for healthier products. However, having a cocktail of interventions at the same time made it hard to disentangle which of all the possible channels was driving the results in this study. Rahkovsky et al. (2013) [32] analyzed the change in customer purchases behavior regarding cereals in a chain of grocery stores that introduced the Guiding Star Program (GSP) three-star ranking system, which includes 0-star unlabeled products, versus similar US grocery stores that did not use the system. They observed that demand increased for more nutritious GSP-ranked cereals and decreased for less nutritious GSP-ranked cereals. However, having unlabeled products and not analyzing other types of products are weaknesses of the GSP system and the study, respectively.

Out of the relevant and most-used classification methodologies (see Annex 1), we selected the Nutri-Score Label, also known as Five-Color Nutrition Label (5-CNL), to inform consumers of the nutritional content of products offered at the university. The Nutri-Score Label meets the requirements for a successful food classification, as established by the Institute of Medicine [39].

Moreover, researchers from the Colombian Ministry of Health and Social Protection were interested in obtaining evidence regarding this label in our context.

The objective of our study was to determine whether randomly providing information to consumers about the meaning of the Nutri-Score labeling system could influence their immediate food-purchasing decision. Specifically, we looked at the intervention effects of randomly providing information and of system awareness between groups on the types of products bought and the likelihood of purchasing products of specific label colors. Moreover, we compared the nutritional quality of purchases between groups.

For our implementation of the Nutri-Score Label, we first piloted the label in a focus group with 6 undergraduate students from different majors. We incorporated their suggestion to use numbers with the colors, instead of the original letters, to correspond to their grade system. In addition, the focus group thought the Nutri-Score Label would be effective in the university population, and believed the label would affect their own purchase behavior if implemented. We translated the Nutri-Score name to Spanish as *Nutripuntos*, but in this paper, we refer to it by its English name.

In addition, to our knowledge, ours is the first paper to study nutrition labeling in Colombia. Our context allows us to focus on the NCD-vulnerable adult population.

### Nutrition Labeling and Information Provision

Nutrition labeling has taken on many forms in attempts to change people’s behavior: point-of-purchase menus (see [40,41,42,43,44,45]) and back-of-package (BOP, see [46,47,48,49,50,51,52]) and front-of-package (FOP) nutrition labeling (see [1,2,18,22,23,24,25,31,32,34,35,36,37,53,54,55,56,57,58]).

Additionally, a review by Seymour et al. (2004) [40] on point-of-purchase environmental interventions observed that most interventions used information strategies. Overall, included studies were observed to increase sales of targeted food or change diets in a positive way. The reviewers also found that these interventions were most effective at worksites and universities, where alternative options were limited.

Focusing on the context of cafeterias, similar to our context, and information provision, Harnack & French (2008) [41] reviewed studies that involved calorie labeling of menus in restaurants and cafeterias. They found that although there was some evidence of reducing calories purchased, the evidence had been inconsistent and weak, and the reviewers called for more rigorous studies in this area. An update to this review, focusing on quick-service restaurants, concluded that calorie labeling on menus did not decrease calorie ordering and consumption in included studies [42]. To note, Wisdom, Downs, & Loewenstein (2010) [59], using a more experimental approach, where participants were given the food they chose without knowing it was still part of the study, found that calorie information significantly reduced caloric intake.

More recently, studies of information provision of calories on menus at full-service sit-down restaurants has had promising results. Ellison, Lusk, & Davis (2014) [60] found that providing calorie information combined with a traffic-light interpretation was able to reduce calories ordered at their restaurant on a university campus. Furthermore, Cawley, Susskind, & Willage (2018) [61], in their working paper, conducted a randomized controlled field experiment using two full-service sit-down restaurants on a university campus. The authors found that providing calorie count information reduced the number of calories ordered.

Our paper focuses on implementing an FOP nutrition label and randomly providing information about it in a university shop that sells food products. As mentioned above, the most relevant and most-used classification methodologies for FOP labels are detailed in Annex 1. From these labeling systems, we chose the Nutri-Score label for our intervention.

Also, as mentioned earlier, the Nutri-Score label meets all Institute of Medicine requirements for successful food classification [39]. In summary: this system does not require specific nutrition knowledge to understand it, it serves as guidance rather than giving specific data, it offers information using a ranking, it uses symbols that are identifiable and easy to remember, is based on the most recent dietary guidelines and consensus reports, and it promotes product comparison and gives the choice of selecting healthier foods by consumers at the point of purchase.

In addition to meeting these requirements, the application of the Nutri-Score label is simple and makes the purchase decision-making process easier. Furthermore, in an online virtual supermarket, the Nutri-Score label was better able to encourage the selection of items with higher nutritional quality than four other types of nutrition labels [23]. In addition, in experimental labs located in urban areas in France, when given the Nutri-Score label and information about the label, participants selected items with higher nutritional quality for sweet biscuits [22].

Furthermore, in a real-world setting, an intervention in Boston, USA found an FOP overall traffic-light system to be effective in changing participants’ health awareness and in prompting them to make healthier food choices at the point of purchase [36,37,55]. The Nutri-Score Label extends from this type of traffic-light system.

The Nutri-Score label is based on the computation of the UK Food Standards Agency (FSA) nutrient profiling system, and considers several types of information: calories, simple sugar, saturated fatty acids, sodium, fiber, protein, and the percentage of fruits and vegetables per 100 g of product. The system summarizes this information into a single indicator of the nutritional quality of the food (nutrition score) and the product is assigned one of the five color categories and, in our case, its respective number, according to the method’s score cutoff values [23] (See [23] for full classification procedure).

## 2. Materials and Methods

### 2.1. Study Context

The experiment took place in one university campus cafeteria, managed by an office called Food Services (*Servicios de Alimentación*) of Pontificia Universidad Javeriana in Bogotá. Food Services also manages other 33 locations, scattered around the campus. Food Services itself uses some of these locations as cafeterias (20) and some as restaurants (4), but rents the remaining ones out to third-party chains (9). Food Services manages the products and workers at their cafeterias and restaurants, and keeps an organized record of every purchase made. Profits from the stores are used for scholarships for underprivileged students. The employees from the cafeteria where the experiment took place had to retrieve products for costumers.

There are several reasons why we chose to implement the Nutri-Score label in partnership with Food Services. First, the shops are an ideal setting to study the effect of FOP labels on consumer spending because of the level of organization in tracking purchases. Food Services uses a centralized inventory and sales system, which keeps a record of every purchase. The computerized system tracks information on price, quantity, and item bought. In addition, every item has a unique code and cashiers must register every purchase. Second, we had access to the recipes for baked goods, which allowed us to extract nutritional information. Finally, all baked goods available for purchase were already using small paper tags containing the name and price of each product. These factors allowed us to carry out the experiment as described below.

The cafeteria where the experiment took place did not differ significantly from the other 19 cafeterias on campus in any of the characteristics we measured through a pilot survey (number of products, types of products, number of employees, tables, chairs, and how products were displayed).

The School of Economics and Business Administration Research Ethics Committee of Pontificia Universidad Javeriana approved this research project on 2 May 2017, Note No. FCEA-DF-0117-2017, before the experiment began. The research project is filed under code 7816. Our project did not work with any vulnerable individuals and did not offer remuneration for participation in the study.

### 2.2. Labeling Products

Food Services offers items prepared on-site, bakery products, and pre-packaged products. We labeled bakery and pre-packaged products, and not those prepared on-site. Bakery products included items without a specific brand such as *alfajores*, *almojabanas* and *pandebono*, which were either manufactured by Food Services in a facility different from the store where the experiment took place, or bought from a third party without any packaging or specific brand. Pre-packaged goods included commercial products such as bottled water, soft drinks and juices, packaged cookies and chips, yogurts, candies, and chocolates, sold under a specific brand. Pre-packaged goods also included fruits, fruit salads and green salads. We did not label items cooked or prepared on-site such as hot chocolate, tea, coffee and ice cream scoops. Our reasoning for this was that pre-packaged food and bakery products have more standardized portions of ingredients than items prepared on-site.

To find out which products needed to be labeled, we first obtained an administrative dataset on sales at the intervention store by item during May 2017—data between August 2016 and July 2017 was also available, but May 2017 was the latest month that fit within the academic semester. During this month, 578 different items were purchased at our intervention store. After filtering out items prepared on-site, we were left with 313 items that needed to be labeled. From the 313, we could not retrieve nutritional information on 13 items (4.2%), because we could not find recipes (*n* = 7) or they were discontinued (*n* = 6). From the remaining 300 items that were successfully labeled, 44 items (14.1%) were labeled as Green, 33 (10.5%) as Light Green, 36 (11.5%) as Orange, 75 (24.0%) as Pink, and 112 (35.8%) as Red. 

To classify commercial products according to the Nutri-Score algorithm (Ducrot et al., 2016) [23], we used the nutritional information on the back of the package from the product packaging itself. To classify bakery products and green salads, we obtained the recipe for each product from Food Services, including ingredients and portions. To translate the recipes into nutritional information, we used a nutritional table published by the WHO for ingredients found in Central America, Latin America, and the Caribbean [62]. In addition, according to the algorithm, fruits and fruit salads got a Nutri-Score of 5—Green or healthiest. 

### 2.3. Intervention Materials

The main intervention materials comprised two types of laminated information placards. Figure 1a shows our control placard (http://www.bancomundial.org/es/news/feature/2018/01/10/comer-fuera-para-muchos-significa-ir-a-un-puesto-de-comida-callejera), and Figure 1b shows the treatment placard. The control and treatment placards contained a news article related to processed food consumption in Latin American countries. We chose a news article related to nutrition in order to make control and treatment placards as comparable as possible. The reason why we gave the same news article to both treatment and control groups was to control the content of the news article itself. If there were any effect of the news article over consumption decisions, then it would affect both groups equally, meaning any differences we found between groups would be in addition to the effect of the news article. The difference between the treatment and control placard was that the information on the Nutri-Score labeling system existence was present only on the treatment placard.

For baked goods, we took advantage of the price tags (see Figure 2). We modified the background color of the original price tags (Figure 2a) and added a five-color scale with numbers—from one to five—at the top of the tag (Figure 2b). The assumption was that it would be harder (or even impossible) for people with the control placard than for people with the treatment placard to infer that the colors and numbers at the top of the price tag were related to a nutritional scale. According to the survey, 83% of individuals in the control group responded that they did not know what the Nutri-Score system was.

For pre-packaged items, we used Nutri-Score stickers for sticking onto pre-packaged products (Figure 2c). These stickers contained only the color and number scale that changed according to its Nutri-Score classification.

In the early morning before the store opened, we labeled all products in the store with Nutri-Score labels. We put the sticker labels directly on the first item on a shelf for pre-packaged products, since these products were displayed in rows; in addition, we put the modified price tags inside the stands in front of their respective baked goods. We also instructed the employees at the shop not to tell consumers the purpose of the system when asked, and instead, if asked, to tell them it was something for internal use. Stickers and price tags were constantly monitored to ensure their permanence and that they were correctly displayed.

After all Nutri-Score labels were in place, we started the intervention. We applied the experiment for up to 9 hours per day during the weekdays of a four-week period (26 February to 23 March 2018). We avoided meal hours because we were concerned that behaviors may be different, and because the flow of customers peaked during these hours.

### 2.4. Experiment Design

Our experiment consisted of randomly informing people about the existence of a new labeling system called Nutri-Score. To do so, we randomly handed control or treatment placards to customers in line, in 10-minute time slots, for the duration. During this 10-minute window we switched between giving the treatment information placards or the control information placards to the experiment subjects. The placards were given to an individual when he or she came alone or to each member of a group at the end of the purchase line. Participants were not recruited beforehand. They were normal customers in the purchasing line waiting to be attended to. Research assistants were instructed not to give any additional information after handing out the placards, and to move away from the participant after the placard was given. We did this without any indication that we were observing their subsequent behavior.

After a participant’s purchase or a group of participants’ purchases, we asked them for their receipt (making sure no one currently reading the placards in line saw this). We then invited the participants to fill out a questionnaire, with the assistance of the experimenters, which took about 10 minutes each. The questionnaire was applied in an area hidden from the main queue. This was to reduce the experimenter expectancy effect [63].

### 2.5. Specific Objectives:

1. To characterize treated and control participants of our Nutri-Score intervention.

2. To quantify the effect of randomly providing Nutri-Score information on total money spent by participants as well as on money spent by label color (green, light green, orange, pink and red).

3. To quantify the effect of randomly providing Nutri-Score information on the likelihood of purchasing at least one item of a specific label color.

4. To quantify whether the nutritional quality of items purchased between treated and control participants is different in terms of calories, protein, saturated fat, fiber, sodium and sugar.

### 2.6. Study Outcomes

Our data comes from the receipts and questionnaires obtained from participants. The receipts told us which products were bought and the purchase price of each product. Unfortunately, we cannot necessarily infer if the participant ate everything they bought. See Figure A1 for an example of a receipt. Notice that one purchase (receipt) can include more than one item, or several units of the same item. A person usually buys several food items per purchase—for example, one green item and two red items. This is why we measured buying *at least one* green (red, etc.) item. For each purchase, we constructed a list of items bought, and then linked each item to its nutritional information and the Nutri-Score categories. This is how we were able to determine the amount of specific nutrients bought in each purchase. In addition, we were also able to collect the price paid per item and total sales per transaction.

The questionnaire included anthropometric and demographic characteristics, usage of the labeling system, and physical activity measurements using the short-form International Physical Activity Questionnaire (IPAQ, [64,65]). These characteristics were later used as controls in regressions.

We applied one questionnaire per subject. Also notice that each subject participated in our experiment only once. As a result, we had one receipt per customer. That is why outcomes were measured at the consumer level.

### 2.7. Statistical Analyses

We calculated the following regression,
$$y_{i}=\alpha +\beta A_{i}+\gamma X_{i}+\varepsilon_{i}$$
where *y_i_* is the outcome of interest of customer (or receipt) *i*, *A_i_* is an indicator variable for random assignment to the treatment group, and *X_i_* is a vector of controls at the individual level that includes age, sex and GPA. We also control for BMI category (underweight, normal, overweight and obese), IPAQ physical activity category (high, medium and low), and socioeconomic category measured by *estrato* (1 through 6). Other covariates include number of days with breakfast in a typical week, alcohol drinker or not, smoker or not and if the consumer usually observes nutrition labels or not. We also control for indicator variables for day of the week, because people may behave differently on certain days, and week of the year, to address any seasonality in decision-making. The variable $\varepsilon_{i}$ is an error term. We use robust standard errors. We estimate the equation using ordinary least squares for continuous outcomes, and a Linear Probability Model (LPM) for binary outcomes. As a robust check for binary outcomes, we also use a Probit Model, a non-linear probability model.

The main outcome of interest is nominal expenditure (overall, and on items of a given color classification). We also analyze the extensive and intensive margins of nominal expenditure on items of a given color classification; i.e., whether the customer ordered at least one item of a given color and how much money the customer spent on items of that color conditional on buying at least one. The last outcome is the nutritional content per purchase (calories, saturated fat, fiber, protein, sodium and sugar).

The parameter *β* is an intention to treat (ITT) effect, i.e., the average treatment effect of being randomly assigned to the treatment group and randomly being provided Nutri-Score Label information. However, we also estimate a local average treatment effect (LATE) below. The differences are explained below.

## 3. Results

We collected 490 receipts; however 2 people refused to take the after-purchase survey. Three people (2 controls and 1 treated) out of the 488 refused to give information on height and weight. Hence, we had valid data on 485 receipts and surveys: 228 control and 257 treated. However, 73 people, 40 controls and 33 treated did not have a GPA (they were freshmen or not university students). To keep these observations, we used a dummy variable as a control when the GPA was missing a value (see Appendix B.2).

Table 1 presents descriptive statistics of treated and control subjects. Panel A shows that our randomization procedure of 10-minute slots gave us control and treatment groups that were well balanced in their characteristics: differences were not statistically different from zero. The groups were similar in terms of several individual pre-determined characteristics and whether they normally use the “Nutrition Facts” on the back-of- pack.

Panel B shows that 83% of control individuals did not know what the Nutri-Score system was (system knowledge), versus 21% of treated individuals. And 21% of treated individuals mentioned using the system when deciding what to purchase (system usage) versus 4% of controls. We will study this further below, when comparing ITT and LATE estimates.

We start by analyzing intention to treat (ITT) estimates—i.e., the effect of being randomly assigned to the treatment group and hence randomly receiving the treatment placard with the Nutri-Score Label information—over expenditure measured US dollars (US $, but we only use the $ sign). Figure 3 shows the unconditional mean expenditure by treated and control groups, and the 95% confidence interval. Mean expenditure was significantly higher for the treatment group than the control group ($1.40 vs $1.59). Figure 4 shows unconditional mean expenditure by color of the item purchased by treated and control groups, and the 95% confidence interval. Expenditure on green items was higher among treated subjects ($0.33 vs $0.52), with no difference in expenditure on the other colors. We further study these results using regressions in Tables 3 and 4.

Table 2 Column (1) shows that randomly providing Nutri-Score Label information increased total expenditure by $0.18 (significant at the 5% level). Columns (2) to (7) provide further results looking at money spent on each color separately. The overall result is that the additional money was mostly spent on healthier items: money spent on green items increased by $0.20 (significant at the 1% level). Also, consumption of non-labeled products decreased by $0.09. In a separate regression, we study the effect of the intervention on the natural logarithm of total money spent. This regression shows that the intervention increased consumers’ purchases by 11.4% (see Appendix B.2). Table A1 provides robustness results of regressions using the same baseline sample but with no controls. As seen, the results do not change much in response to the chosen covariates, as expected.

So far, we have estimated an ITT. We now move to the estimation of the local average treatment effect (LATE)—i.e., an average treatment effect on the compliant subpopulation of treated subjects. We interpret LATE as the effect of being randomly aware of the system. The main difference between ITT and LATE is that not everyone assigned to the treatment group may actually have been exposed to the treatment (seen the placard); some people may have ordered without looking at the placard or may have looked at the placard but not noticed the Nutri-Score system. To investigate this, the questionnaire had a question asking whether subjects used the labeling system, or if he or she did not know about the system. We asked the following question: “*Did you use the new labeling system for food and drink products in deciding what to purchase?*” Subjects then chose one of three options: (a) I don’t know what it is, (b) no and (c) yes (see Appendix B.2). We classified someone as aware of the system existence if they answered “yes” or “no” to this question. If a person answered, “I don’t know what it is”, then we classify that person as not being aware of the system’s existence. According to Table 1, the majority of the treatment group, 79%, were aware of the Nutri-Score system, but 17% of the control group were as well.

To estimate the LATE, we first estimated a model of awareness of the system as a function of random assignment to the treatment group and the full set of controls, and found that random assignment to the treatment group was associated with a 61.5 percentage point increase in the probability of reporting knowing that the system exists (see Appendix B.2). To estimate a LATE, we follow Imbens & Angrist (1994) [66] and used assignment to treatment as an instrumental variable for system awareness. The first stage is the model of awareness of the system just described.

Row B of Table 3 shows LATE estimates. Column (1) shows that being aware of the system increased expenditure by $0.29, much higher than the ITT estimate (the effect of providing information) of $0.18. The difference lies in the non-compliant subpopulation, those given the placard but did not look at it or did not learn about the Nutri-Score system. Had these people learned about the system, the effect would have been greater. The LATE increase is due to a $0.33 increased expenditure on green items, and a $0.15 decreased expenditure on non-labeled items.

The main takeaway is that randomly providing information on the Nutri-Score Label increased expenditure by $0.18, but being aware of the Nutri-Score Label increase expenditure by $0.29.

We now move on to analyzing the extensive and intensive margins by classification color; the extensive margin is the probability of buying any item of a given color (green, light green, etc.) and the intensive margin is the amount of money spent on items in that color category conditional on buying any items of that color. Note that a person can buy one item (or many items) of a given color or none at all. We start with the intensive margin in Table 4. Column (1) shows that, conditional on buying at least one green item, the intervention increased expenditure on green items by $0.34. For this analysis, there is no statistical evidence that expenditure decreased on the least healthy items (i.e., red and pink), or non-labeled items. Table A2 shows robustness results for the effect of the intervention on total money spent and on the natural logarithm of total money spent, with and without covariates.

The previous ITT and LATE results showed that expenditure on healthier items increased and expenditure on non-labeled items decreased. However, this result is consistent with an unchanged demand curve for non-labeled items, if people are buying less expensive non-labeled items. We now study what happened with the demand, or the likelihood of purchasing an item of a given color, i.e., the extensive margin.

Figure 5 shows the fraction of items purchased of each color with 95% confidence interval error bars. The fraction of green items in an average purchase was higher among customers assigned to the treated group than among customers assigned to the control group. We study this further in Table 5 using regressions.

Table 5 shows different estimates of the extensive margin: the likelihood of purchasing at least one item, by color, using the full set of controls, and a linear probability model. Row A of Table 5 shows ITT estimates. Column (1) shows that randomly giving the Nutri-Score system information made customers 0.10 percentage points more likely to buy at least one green item; Column (6) shows that customers were 0.09 percentage points less likely to buy at least one non-labeled product. Both were significant at the 5% level. Row B shows LATE on the compliant subpopulation of successfully “treated” subjects. The estimates on Columns (1) and (6) shows that being aware of the Nutri-Score system made consumers 0.16 percentage points more likely to buy at least one green item and 0.15 percentage points less likely to buy at least one non-labeled item, both significant at the 5% level. Table A3 shows additional results a non-linear Probit Model.

Altogether, neither providing Nutri-Score Label information nor being aware of the Nutri-Score Label changed the probability of buying the least healthy items (i.e., pink and red).

The shift in demand also changed the nutritional intake. Figure 6 shows the nutritional content of bought items, by nutrient and randomization group with a standardized serving size of 100 grams for solid food or 100 milliliters for beverages. The treated group purchased on average more protein than the control group. Figure A2 uses a non-standardized serving size. We study this further in Table 6 using regressions.

Table 6 presents regression results for ITT and LATE estimates, using the full set of controls with a standardized serving size of 100 grams for solid food or 100 milliliters for beverages. Row A shows that randomly providing Nutri-Score Label information increased the protein content of purchases by 2.73 grams. Row B shows that being aware of the system increased protein purchase by 4.31 grams. There was no statistically significant change in the purchase of other nutritional content. However, the point estimate in Column (2) shows that calorie content of purchases increased, which was not a desirable outcome, in addition to purchases of unhealthful items not decreasing. Therefore, providing information to customers about the meaning of the Nutri-Score Label does not necessarily decrease the calorie content of their purchases.

## 4. Discussion

Our overall results indicate that those in the treated group bought similar items to the control group and at least one additional green-labeled product, which resulted in those in the treated group buying items containing more protein. Since both treatment and control groups were given the same news article, we do not believe these results are due to the content of the news article and these results would be different if our treatment were instead supplemented with different information. Below we organize our discussion by our specific objectives.

### 4.1. Characterize Treated and Control Participants of Our Nutri-Score Intervention

Treated and control groups were well balanced in all important and relevant characteristics. This allowed us to attribute any differences in purchasing decisions between the groups to our intervention treatment. Furthermore, as expected and as an important check for our intervention, the treated group reported both greater knowledge and usage of the Nutri-Score label.

### 4.2. Quantify the Effect of Randomly Providing Nutri-Score Information on Total Money Spent by Participants as well as on Money Spent by Label Color (Green, Light Green, Orange, Pink and Red)

Those in the treated group had a higher total expenditure compared to controls. This was driven by higher expenditure on green-labeled products. Interestingly, expenditures in products labeled with other colors were not different between treatment and control. This is an analysis that has been lacking in many of the previous studies. These results are in line with the single summary traffic-light label intervention in hospital cafeterias in Boston with participants buying healthier products [36,37,55].

It is also noteworthy that participants spent their own money and were not given any money, credit, vouchers, or reimbursements beforehand or afterwards. This analysis is important for stores, since they usually need to make a profit or at least not lose money, as in our university context.

### 4.3. Quantify the Effect of Randomly Providing Nutri-Score Information on the Likelihood of Purchasing at least One Item of a Specific Label Color

Those in the treated group had a higher probability of buying at least one green-labeled (most nutritious) product. This meant that those exposed to the treatment information were more likely to buy green-labeled products immediately after this exposure. This was true whether we used ITT analyses, which only analyzes the effect of randomization into treatment groups, or LATE analyses, which take into account that some treated participants did not look at or understand the information on the placard. This is another demonstration of the treatment effect, again in line with the Boston hospital cafeteria studies [36,37,55].

However, there were no differences between treatment and control in probability of buying at least one product labeled with the other less healthy colors.

### 4.4. Quantify Whether the Nutritional Quality of Items Purchased Between Treated and Control Participants is Different in Terms of Calories, Protein, Saturated Fat, Fiber, Sodium and Sugar

Treated customers bought products with higher concentrations of protein than control customers did. This was due to the fact that the green-labeled products available contained a higher concentration of protein overall than items labeled with other colors. This result is in line with what was found in the Ducrot et al. (2016) [23] online supermarket study as well as the Julia et al. (2016) [22] experimental lab store study in the label plus information arm for sweet biscuits, where the Nutri-Score label promoted higher nutrition quality in selected items for participants of both studies.

Together with the above results, this leads us to believe that people from the treated group bought similar items to the control group, and at least one green item in addition.

However, treated customers purchased more calorie content compared to control customers. This result was not desirable and is contrary to the results of Cawley et al.; Ellison et al.; and Wisdom et al. [59,60,61], who all found that providing calorie count information (for Ellison et al. [60] only in conjunction with a traffic light interpretation) decreased calories purchased. This, coupled with the fact that there were no differences between groups in the likelihood of purchasing less healthy items, gives us pause. There is a need for more research before implementing any policy with the Nutri-Score label along with information provision about the label, in our context.

Although we did not look at actual consumption, treated customers may have then eaten less of the less healthy item they bought, as was in the case of a school cafeteria convenience-line intervention, where those who bought healthier items consumed less of the unhealthier items they bought [67].

We found an immediate effect of being exposed to Nutri-Score system information, which over time and multiple exposures, could result in more subjects understanding the system and more of them choosing to use the system for their food choices. Therefore, in a larger, longer-term campaign, with exposure to posters, traditional media, and social media containing information about the system coupled with wider Nutri-Score system implementation, we would predict more system awareness and more system use at the point of purchase. However, this remains to be seen.

Our current study accounts for methodological deficiencies in other nutrition labeling studies including many studies in point-of-purchase calorie labeling [41,42]. These include a field experiment in a real-world setting, looking at real purchase behavior, and using a control group [17,18,19].

In terms of the customers that were aware of the Nutri-Score system but did not use it for choosing their purchase, application to our context of the Burton & Kees framework [68], developed for calorie labeling, may shed some light on this behavior. According to the framework, these customers may not have been motivated to use the system for individual reasons, such as not wanting to spend more money on healthier items. Alternatively, they may have had certain price calorie expectations, which they may have assumed the Nutri-Score system would not help them with, if they wanted to be full. Or they may have already determined what they wanted to buy beforehand, so with no choice to be made, they had no use for the Nutri-Score system, this time [68].

It is worth noting that our results may be context specific, at the university, where alternative options are limited, which has been shown to be one of the contexts where point-of-purchase interventions has the greatest effect [40].

In addition, to note, the drop in expenditure for non-labeled products for the treated group may be because of a lack of signaling of “quality” in this case, nutritional quality, of these products. Customers may then interpret this as the products may be of low nutritional quality, following the model of Mojduszka & Caswell [69], and therefore be wary of buying them.

### 4.5. Limitations

Our results are limited to those in a university community and may not generalize well. Most of our subjects were students with an average age of 19, which may limit generalizability. However, the university consists of adults, and adults are at the highest risk of overweight and obesity and they have the autonomy to change their purchase behaviors. In addition, green items on our campus are more expensive than those outside the university context, so ours could be lower bound results and in stores outside the university the differences in purchases could be more pronounced, meaning more green-labeled items may be bought if this was outside the university context.

In addition, our study looked at purchases but not the actual consumption of the foods. It may be the case that customers bought extra healthier items, but then threw them away, as occurred in a short-term school convenience-line intervention [67], which could be a problem if the system is implemented, at least at first, in terms of food waste. The authors believed, however, that additional exposure to the healthier food on students’ trays would eventually result in increased consumption of the healthier food as well.

### 4.6. Further Research

Studying the long-term effects of the information and system may provide further insight. Moreover, combining this labeling intervention with other types of interventions, such as taxes on unhealthy products or subsidies on healthy products, may make the system even more effective. These hypotheses have yet to be explored.

## 5. Conclusions

Our intervention was one of giving information on the Nutri-Score label. In summary, there was an increased demand for green products reflected in (i) a higher probability of buying at least one green-labeled (most nutritious) product and (ii) a higher total expenditure, concentrated in green-labeled products for the treated customers. In addition, levels of protein purchased among treated customers were higher than those in the control group. Interestingly, expenditures in items labeled with the other colors were not different between treatment and control. This leads us to believe that people from the treated group bought similar items to the control group and a green-labeled product in addition.

In our sample, providing information about the Nutri-Score label was effective in changing the immediate purchase behavior of customers at a store on a university campus. This immediate effect is already significant and promising. The effect of being aware of the Nutri-Score Label increased total expenditure, expenditure on green-labeled products, the likelihood of buying at least one green-labeled item, and the concentration of protein in items bought, in an amount greater than the effect of information.

Furthermore, the augmentation of profit for the store is a positive result. From our data, when implementing the Nutri-Score label at a store in our context, while helping customers make healthier choices, the store also made more money at the same time from treated customers. This result may ease the implementation of such front-of-package nutrition labels in our context by giving a potential financial incentive to stores for implementing nutrition labels such as the Nutri-Score label.

## Figures and Tables

**Figure 1 nutrients-11-00491-f001:**
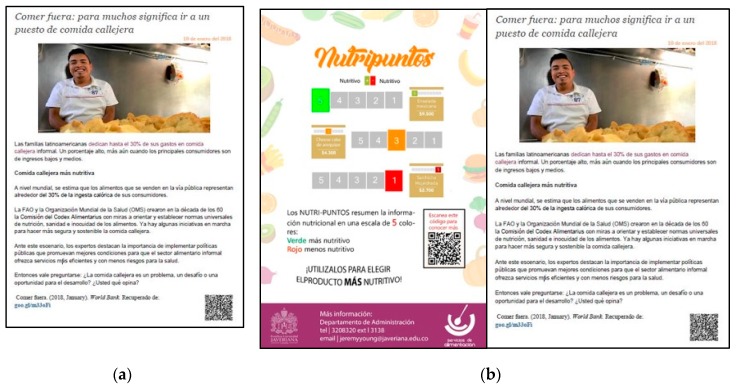
Information placards for control and treatment groups. (**a**) control placard; (**b**) treatment placard.

**Figure 2 nutrients-11-00491-f002:**
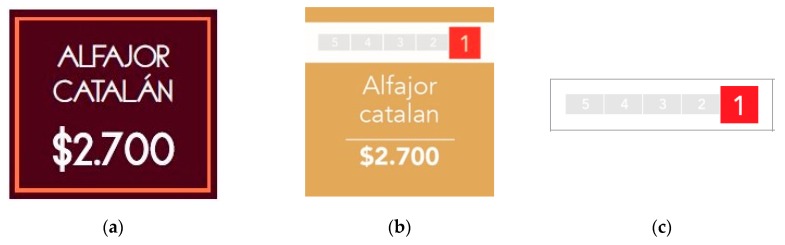
Price tags inside the cafeteria stands before and during intervention. (**a**) Price tags before intervention; (**b**) modified price tags during intervention, which have a color and number scale that changes depending on the nutritional information of the product; (**c**) stickers for sticking onto pre-packaged products, which also have a color and number scale that changes depending on the nutritional information of the product. Notice that there is no reference to the Nutri-Score system in any of the price tags. Figures are not to scale.

**Figure 3 nutrients-11-00491-f003:**
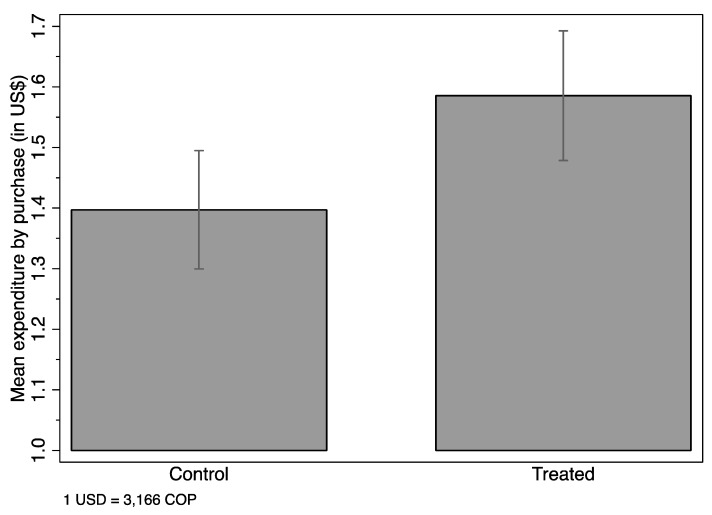
Mean expenditure by purchase of treated and control groups. Notes: This figure shows total expenditure regardless of the item’s color, among control and treated subjects, in USD. The vertical whiskers refer to the 95% confidence interval. The vertical axis starts at $1.

**Figure 4 nutrients-11-00491-f004:**
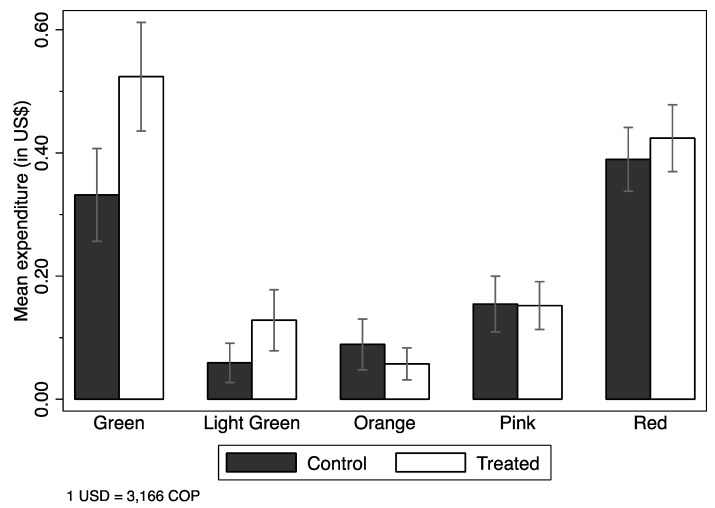
Unconditional mean expenditure of items per purchase, by Nutri-Score classification, of treated and control groups.

**Figure 5 nutrients-11-00491-f005:**
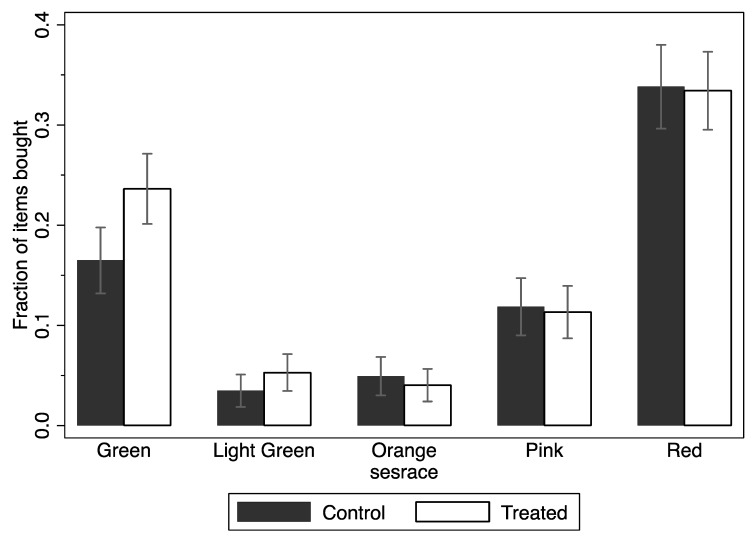
Fraction of items bought by color composition and randomization group.

**Figure 6 nutrients-11-00491-f006:**
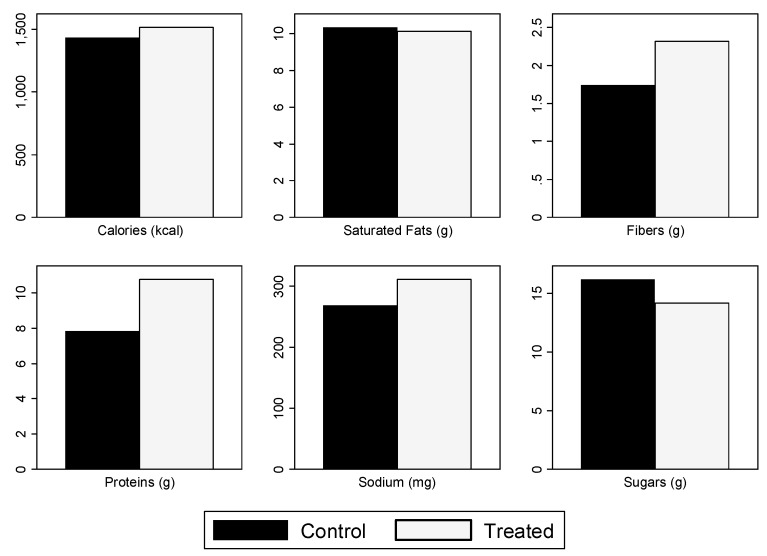
Nutritional content of purchases, by nutrient and randomization group, using a standardized serving size of 100 g or mL.

**Table 1 nutrients-11-00491-t001:** Balanced characteristics between control and treated groups.

	Control		Treated		Difference in Mean (1)–(3)		
	Mean	*N*	Mean	*N*		*p*-Value	*N*
	(1)	(2)	(3)	(4)	(5)	(6)	(7)
**A. Characteristics**							
Days of week with breakfast	5.99	230	5.67	258	0.32	0.06	488
Height (cm)	166.80	228	167.77	257	−0.98	0.25	485
Weight (kg)	60.37	228	62.17	257	−1.80	0.08	485
BMI	21.60	228	21.99	257	−0.39	0.15	485
GPA (out of 4)	3.99	188	3.97	224	0.02	0.55	412
Age	19.50	229	19.59	258	−0.09	0.73	487
Socioeconomic neighborhood (*estrato*)	4.12	229	4.06	258	0.06	0.55	487
High physical activity (IPAQ)	0.36	230	0.44	258	−0.08	0.07	488
Medium physical activity (IPAQ)	0.29	230	0.20	258	0.09	0.02	488
Low physical activity (IPAQ)	0.35	230	0.36	258	−0.01	0.85	488
Student	0.97	230	0.97	258	0.00	0.75	488
In a group	0.39	230	0.49	258	−0.11	0.02	488
“Nutrition Facts” use to purchase	0.43	230	0.50	258	−0.07	0.12	488
Smoker	0.20	230	0.19	258	0.01	0.87	488
Drinker	0.48	230	0.48	258	0.00	0.97	488
Visit other cafeterias	0.80	229	0.90	258	−0.10	0.00	487
Women	0.59	230	0.56	258	0.03	0.51	488
Underweight (BMI < 18.5)	0.12	230	0.09	258	0.03	0.31	488
Normal (BMI ≥ 18.5 & BMI < 25)	0.78	230	0.76	258	0.02	0.63	488
Overweight (BMI ≥ 25.0 & BMI < 30)	0.08	230	0.13	258	−0.05	0.07	488
Obese (BMI ≥ 30.0)	0.03	230	0.02	258	0.00	0.84	488
**B. Nutri-Score System Usage**							
Used Nutri-Score system	0.04	230	0.21	258	−0.17	0.00	488
Did not use Nutri-Score system	0.13	230	0.58	258	−0.44	0.00	488
Did not know Nutri-Score system	0.83	230	0.21	258	0.62	0.00	488

**Table 2 nutrients-11-00491-t002:** Unconditional effect of providing Nutri-Score Label information on total expenditure and expenditure by color. Intention to treat (ITT) estimates.

	Total	Green	Light Green	Orange	Pink	Red	Non-Labeled
	(1)	(2)	(3)	(4)	(5)	(6)	(7)
Estimated Effect	0.179 **	0.201 ***	0.058	−0.037	0.009	0.041	−0.093 **
	{0.078}	{0.075}	{0.034}	{0.035}	{0.037}	{0.045}	{0.045}
*N*	484	484	484	484	484	484	484

Notes: Each column shows the results of separate regressions. The dependent variable in Column (1) is the amount of money spent by a consumer on his/her purchase, in Column (2) is the amount of money spent in green items only by a consumer in his/her purchase, etc. If a consumer did not buy any green items, then the amount of money spent is taken as zero. All specifications include the following full set of pre-determined controls: individual controls (GPA, age, freshman, estrato, BMI, breakfast, nutritional facts use, smoker, drinker and sex), day of week and week number. Robust standard errors in parenthesis. ** *p* < 0.05 *** *p* < 0.01.

**Table 3 nutrients-11-00491-t003:** ITT and LATE effects of providing Nutri-Score Label information on total expenditure and expenditure by color, using different estimates.

	Total	Green	Light Green	Orange	Pink	Red	Non-labeled
	(1)	(2)	(3)	(4)	(5)	(6)	(7)
A. Intention to treat (ITT)	0.179 **	0.201 ***	0.058	−0.037	0.009	0.041	−0.093 **
	{0.078}	{0.075}	{0.034}	{0.035}	{0.037}	{0.045}	{0.045}
*N*	484	484	484	484	484	484	484
B. Local average treatment effect (LATE)	0.292 **	0.327 ***	0.095	−0.059	0.014	0.067	−0.152 **
	{0.123}	{0.119}	{0.054}	{0.055}	{0.058}	{0.071}	{0.071}
*N*	484	484	484	484	484	484	484

Notes: This table follows the same specification as Table 2. Robust standard errors in parenthesis. ** *p* < 0.05 *** *p* < 0.01.

**Table 4 nutrients-11-00491-t004:** Effect of providing Nutri-Score Label information on expenditure by color, conditional on buying at least one item of a given color.

	Green	Light Green	Orange	Pink	Red	Non-Labeled
	(1)	(2)	(3)	(4)	(5)	(6)
Estimated Effect	0.341 **	0.483	−1.698	−0.043	0.047	−0.047
	{0.146}	{0.478}	{0.995]	{0.111}	{0.048}	{0.051}
*N*	137	33	32	83	223	179

Notes: The dependent variable in this table is the amount of money spent by each person (or on each purchase) on items of the indicated color in each column, conditional on buying at least one item of the same color. This table uses the same covariates as in Table 2. Robust standard errors in parenthesis. ** *p* < 0.05.

**Table 5 nutrients-11-00491-t005:** ITT and LATE effects of providing Nutri-Score Label information on the probability of purchasing at least one item of a particular color using a linear probability model (LPM).

	Green	Light Green	Orange	Pink	Red	Non-Labeled
	(1)	(2)	(3)	(4)	(5)	(6)
A. Intention to Treat (ITT)	0.096 **	0.025	−0.012	−0.004	0.021	−0.093 **
	{0.042}	{0.023}	{0.025}	{0.035]	{0.046}	{0.045}
*N*	484	484	484	484	484	484
B. Local average treatment effect (LATE)	0.156 **	0.041	−0.020	−0.007	0.034	−0.151 **
	{0.067}	{0.036}	{0.039}	{0.055}	{0.072}	{0.072}
*N*	484	484	484	484	484	484

Notes: The dependent variable in this table is a dummy that takes the value of one if the person bought at least one item of the color indicated in each column, and zero otherwise. All other specifications are the same as in Table 2. Robust standard errors in parenthesis. ** *p* < 0.05.

**Table 6 nutrients-11-00491-t006:** ITT and LATE effects of providing Nutri-Score Label information over nutritional content of bought items in a standardized serving size (100 g for solid or 100 mL for liquid), by nutrient.

	Protein (g)	Calories (kcal)	Sugar (g)	Sodium (mg)	Saturated Fats (g)	Fibers (g)
	(1)	(2)	(3)	(4)	(5)	(6)
A. Intention to Treat (ITT)	2.725 ***	57.027	−2.865	37.668	0.150	0.627
	{0.864}	{113.509}	{2.544}	{25.264}	{1.225}	{0.580}
*N*	393	393	393	393	393	393
B. Local average treatment effect (LATE)	4.311 ***	90.209	−4.532	59.585	0.237	0.992
	{1.339}	{173.514}	{3.915}	{38.642}	{1.872}	{0.889}
*N*	393	393	393	393	393	393

Notes: Robust standard errors in parenthesis. *** *p* < 0.01.

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
