# Peer review of "The Effect of Randomly Providing Nutri-Score Information on Actual Purchases in Colombia"

_nutrients, 2019, doi:10.3390/nu11030491_

Reviewer 1 Report

First of all, I find the research topic important and would like to see the body of knowledge in Nutrition labeling increase by the results of the authors. However, I feel that the current manuscript is not sufficient to make a valuable contribution. Consequently, what the authors will find below are a large number of issues and concerns that need to be addressed to improve the manuscript (from my point of view).

In addition to content, the authors should definitely employ a professional English native speaker to proof read their manuscript (and perhaps also consistently format the manuscript)

1. Introduction:

1.1 Relevance and Motivation of the research start ok and then loose focus. For example, the transition to FOP is quite sudden in Line 42.

1.2 Problem with “experimenter bias” somehow holds for own research.

1.3 Missing papers in Footnote 4: Sacks et al. (2011); Cawley et al. (2015); Boztug et al. (2015); Nikolova & Inman (2015); Elshiewy et al. (2016); Elshiewy & Boztug (2018).

1.4 In addition, see reviews by Hersey et al. (2013), Andrews et al. (2014), Van Kleef & Dagevos (2015) for further input to the topic.

1.5 Literature review in Introduction falls to short (I recommend an introduction with only brief statement about the research gap and an additional chapter with a more comprehensive literature review, especially the studies by Thorndike and colleges are highly relevant to your research. But also the following reviews for menu and restaurant-labeling: see Harnack and French 2008; Seymour et al. 2004; Swartz et al. 2011.

1.6 With a separate literature review, the entire introduction should be revised to provide the following elements as usually in an introduction: relevance of the topic, whats the problem that you solve, knowledge about problem from the literature, research gap connected to problem, intended contribution of your study to close the research gap!

1.7 then the subsection “our study” as well as “research objectives” should be changed by shifting the relevant content to the “introduction” and the “data description”.

2. Methods:

2.1 the study context is not well described

2.2 the same holds for the experimental design (it seems as if only baked goods were labeled, what distribution of NutriPoints do baked goods have? Or were all products labeled? This becomes relevant later in the interpretation of the results)

2.3 the claims made with respect to a “covert study design” are hard to validate;

2.4 Figure XX mentioned in line 225 is missing

3. Statistical Analysis:

3.1 Why is expenditure the dependent variable? that does not make much sense in a dietary context. Are prices related to healthfulness? if so, then Y is confounded!

3.2 total expenditure or only for baked goods? This is still not 100% clear at this point.

3.3 complying with the treatment can also be the result of so-called recall-bias (= post-hoc rationalization of label use while real reasons for their previously made decision was different, see (Malam et al. 2009).)

 4. Results:

4.1 the way how missing values in GPA are handled looks unfamiliar (is this a valid approach?)

4.2 how about matching the treatment and control samples?

4.3 the claim that adding controls reduces standard errors as well as the one in Footnote 9 is not fully correct nor relevant.

4.4 In line 321, how was this “elasticity” for a dummy variable computed? You should follow Halvorsen

and Palmquist (1980).

4.5 Line 351: “Decrease” in non-label items not mentioned

4.6 How is the awareness of the system related to your controls?

4.7 Why not a logistic regression for the model from Table 5? A linear regression can be substantially biased based on the distribution of the binary Y!

4.8 When it comes to the nutrients, it seems that it was calculated for all purchases (while it still appears that only baked goods were labeled)

4.9 What now becomes apparent is that people buy more calories when treated (not significantly more, but they do at least on average): This may allow the conclusion that the Label only increased purchases of green items (therefore more expenditure) but all other items/colors were unaffected, so that people even bought more after treatment!! This comes close to detrimental effects like the one discussed by Chernov & Chandon (2010), Chernev (2011) -- perhaps it makes sense to estimate a kind of density measure per receipt rather than total values.

 5. Discussion

5.1 this part is too descriptive and should rather pick up what was found in a brief manner and then emphasize the contribution w.r.t. what has to be discussed in the literature review

5.2 the implications are not practical as the label did nothing, while only the Placard was used as treatment, so how will stores make use of this? Distribute Placard? Educate the consumers in another way?

5.3 the results make only sense in the setting you have: Cafeteria, therefore look at the reviews by Harnack and French 2008; Seymour et al. 2004; Swartz et al. 2011 to position your setting and findings.

5.4 make the discussion only based on what you found in your results (i.e. empirical support)

5.4 Additional topics for a discussion:

- If the model is valid, the drop in expenditure for missing might be due to “signaling” see (Mojduszka and Caswell 2000)

- The fact that not everyone responds to the treatment (or makes use of the treatment) might be explained by the framework by Burton and Kees (2012).

 6. Conclusions

6.1 perhaps integrate the conclusions into the discussion.

Recommended References

Sacks, Gary, Kim Tikellis, Lynne Millar, and Boyd Swinburn (2011), “Impact of Traffic-Light Nutrition Information on Online Food Purchases in Australia,” Australian and New Zealand Journal of Public Health, 35 (2), 122–26.

Cawley, John, Matthew J. Sweeney, Jeffery Sobal, David R. Just, Harry M. Kaiser, William D. Schulze, et al. (2015), “The Impact of a Supermarket Nutrition Rating System on Purchases of Nutritious and Less Nutritious Foods,” Public Health Nutrition, 18 (1), 8–14.

Boztug, Yasemin, Hans J. Juhl, Ossama Elshiewy, and Morten B. Jensen (2015), “Consumer Response to Monochrome Guideline Daily Amount Nutrition Labels,” Food Policy, 53, 1–8.

Nikolova, Hristina D., and J. Jeffrey Inman (2015), “Healthy Choice: The Effect of Simplified POS Nutritional Information on Consumer Food Choice Behavior,” Journal of Marketing Research, 52 (6), 817–35.

Elshiewy, Ossama, Steffen Jahn, and Yasemin Boztug (2016), “Seduced by the Label: How the Recommended Serving Size on Nutrition Labels Affects Food Sales,” Journal of the Association for Consumer Research, 1 (1), 104–14.

Elshiewy, O., & Boztug, Y. (2018). When Back of Pack Meets Front of Pack: How Salient and Simplified Nutrition Labels Affect Food Sales in Supermarkets. Journal of Public Policy & Marketing, 37(1), 55-67.

Hersey, James C., Kelly C. Wohlgenant, Joanne E. Arsenault, Katherine M. Kosa, and Mary K. Muth (2013), “Effects of Front-of- Package and Shelf Nutrition Labeling Systems on Consumers,” Nutrition Reviews, 71 (1), 1–14

 Andrews, J. Craig, Chung-Tung J. Lin, Alan S. Levy, and Serena Lo (2014), “Consumer Research Needs from the Food and Drug Administration on Front-of-Package Nutritional Labeling,” Journal of Public Policy & Marketing, 33 (1), 10–16.

Van Kleef, Ellen, and Hans Dagevos (2015), “The Growing Role of Front-of-Pack Nutrition Profile Labelling: A Consumer Perspective on Key Issues and Controversies,” Critical Reviews in Food Science and Nutrition, 55 (3), 291–303.

Harnack, L. J. and S. A. French (2008). Effect of Point-of-Purchase Calorie Labeling on Restaurant and Cafeteria Food Choices: A Review of the Literature. International Journal of Behavioral Nutrition and Physical Activity 5 (51).

 Swartz, J. J., D. Braxton and A. J. Viera (2011). Calorie Menu Labeling on Quick-Service Restaurant Menus: An Updated Systematic Review of the Literature. International Journal of Behavioral Nutrition and Physical Activity 8 (135).

 Seymour, J. D., A. L. Yaroch, M. Serdula, H. M. Blanck, and L. K. Khan (2004). Impact of Nutrition Environmental Interventions on Point-of-Purchase Behavior in Adults: A Review. Preventive Medicine 39 (2), 108-136.

 Malam, S., S. Clegg, S. Kirwan, and S. McGinigal (2009). Comprehension and Use of UK Nutrition Signpost Labelling Schemes. Technical report, British Market Research Bureau.

Halvorsen, Robert, and Raymond Palmquist (1980), “The Interpretation of Dummy Variables in Semilogarithmic Equations,” American Economic Review, 70 (3), 474–75.

 Chernev, A., & Chandon, P. (2010). Calorie estimation biases in consumer choice. Leveraging consumer psychology for effective health communications: The obesity challenge, 104-121.

 Chernev, A. (2011). The dieter's paradox. Journal of Consumer Psychology, 21(2), 178-183.

 Mojduszka, Eliza M., and Julie A. Caswell (2000), “A Test of Nutritional Quality Signaling in Food Markets Prior to Implementation of Mandatory Labeling,” American Journal of Agricultural Economics, 82 (2), 298–309.

Burton, Scot, and Jeremy Kees (2012), “Flies in the Ointment? Addressing Potential Impediments to Population-Based Health Benefits of Restaurant Menu Labeling Initiatives,” Journal of Public Policy & Marketing, 31 (2), 232–39.

Reviewer 2 Report

Abstract. Delete final sentence.

Keywords, please add some.

Lines 155-158, delete this sentence

Table 1. I cannot understand the distribution of BMI groups.

There has been growing interest in recent years in front-of-pack food labels (FOP). This study extends the research to Columbia. Most studies on FOP labels have used a simulated shopping environment, most often using a computer. That methodology has potential flaws as the findings may be different in a real supermarket. The investigators here used more of a real-world situation by testing the effect of a FOP label on actual purchases. However, the study has several notable weaknesses.

Section 2.2. I find it difficult to understand what the investigators actually did. The methodology is not explained clearly.

FOP labels are intended for use over a period of years. Shoppers probably need much time before they make the effort to figure out what FOP labels actually mean and then use the information to change their shopping choices. In this study shoppers were educated using a poster. They were expected to read the poster immediately and follow its directions. This is a weak imitation of a real-world situation. Table 1 provides information on subjects’ awareness of the Nutri-score. Apparently only 21 percent of “treated” subjects were aware of Nutri-Score.

The study was done over a period of 25 days. People typically go to the same store regularly and buy the same product. Is it possible that many students learned about the FOP labels and which products were most healthy early in the study and this changed their later food selections (ie over the final 10 days people bought healthy foods during control periods as they had already learned which foods have a high score)?

Another factor that limits the generalizability of the findings is that the subjects were almost all students and had an average age of about 19. Lines 447-453 should state these limitations clearly.

Lines 309-313. This repeats caption to Table 2. Avoid needless repetition.

Table 2 and line 318 state that p<0.1 was classed as significant. I cannot recall ever seeing that done in another study. Please change this.

Tables 3 and 4. The name Nutri-Score has been changed to 5CNL. Be consistent. In lines 376-378 this occurs in 2 lines.

Table 3 and the text that refers to it could be made clearer.

Line 367 states that sales of unhealthy items fell. This seems to contradict lines 358-359 and Figure 5.

Lines 374-382. It is very difficult to understand how the text relates to the numbers in Table 5.

Table 6 must include units.

The Results section would be improved by being simplified and shortened.

Author Response

Please refer to PDF for replies.

Round  2

Reviewer 1 Report

First of all, I appreciate the effort the authors have put into the revision, and I believe the manuscript can be on a good path to publication. Even though I am still not convinced by a number of your rebuttals, I think that the paper can go further as minor revision. However, I still urge the authors to consider the following comments:

1. The focus of the study is not precise from the introduction/literature review and only becomes clear in your empirical part. You don’t look at nutrition labeling as the introduction and literature review suggests! You look at how providing the placard impacts the purchases when labeling is present. Please make this clear as early as possible (title, abstract & introduction) and also focus your results and discussion section to this setting! This means that your implications cannot be linked to labeling as you currently do, but to information provision in combination with labeling as your empirical setting!

You also need to focus on the previous literature (in a brief and concise manner) that is relevant to your study (treatment = providing information, labeling & cafeteria setting!). Only then, you can be more clear about what you are doing and your contribution.

(By the way Line 73 “Two exceptions that use controls are” is not fully correct. Nikolova & Inman also have a control group comparison as robustness check!).

2. I am not convinced by the way you impute missing values for GPA. First, your statement in Footnote 8 is incorrect, namely “there is no valid approach to treat missing values, but dropping the observations with missing values”. The simplest way is to impute missing values by the average value of the non-missing values* (i.e. mean GPA; in the case there are not systematics = MCAR)! I think using the mean should be fine, as it is just a control variable. How do the results look then? If the results are unaffected by the value of your imputation, then use the mean and exclude that additional dummy variable from your control variables! From what I see in the Reviewer Response (p. 11), your results excluding the observations with missing GPA look substantially different than the ones reported in the paper (significant effects for all groups)! That gives the impression that your imputation (plus that additional dummy variable as control) have a too large and biasing impact on your results.

 *In addition, there is regression-based imputation (i.e., predict missing GPA values from a linear regression with GPA as dependent variable and strong explanatory variables from where GPA values are not missing) and the EM-algorithm proposed by Dempster, Laird, & Rubin (1977) [Maximum likelihood from incomplete data via the EM algorithm. Journal of the Royal Statistical Society]. The latter two approaches are just for your information!

3. The main result is increased spending in the green category and decreased spending in the unlabeled category. Perhaps the first result comes from the content of the news article and would look different if your treatment would be supplemented with different information. For example, what about an article that emphasizes health problems due to high sugar consumptions? Would that perhaps lead to a decrease in the red category? Please interpret your results in this light and also discuss this issue (also in your limitations).

4.Your results suggest more calorie intake by the treatment! Be more clear that it is not a desirable outcome and that purchases of unhealthful items did not decrease, and what this means in particular to public policy when it comes to considering your concrete treatment (information provision!).

5. Please consider to employ a professional English-proofreader, as the manuscript still has a lot of grammatical errors.

Author Response

Response to Reviewer 1 Comments

First of all, I appreciate the effort the authors have put into the revision, and I believe the manuscript can be on a good path to publication. Even though I am still not convinced by a number of your rebuttals, I think that the paper can go further as minor revision. However, I still urge the authors to consider the following comments:

Thank you again for the useful comments and suggestions. We have tried to address and incorporate all your comments and suggestions within the time window given.

Point 1.1: The focus of the study is not precise from the introduction/literature review and only becomes clear in your empirical part. You don’t look at nutrition labeling as the introduction and literature review suggests! You look at how providing the placard impacts the purchases when labeling is present. Please make this clear as early as possible (title, abstract & introduction) and also focus your results and discussion section to this setting! This means that your implications cannot be linked to labeling as you currently do, but to information provision in combination with labeling as your empirical setting!

Response 1.1: We agree. ITT results should be interpreted as the effect of randomly providing Nutri-Score Label information. However, LATE results can be interpreted as the effect of being aware of the Nutri-Score Label.

 In any case, the title has been changed to:

 The effect of randomly providing Nutri-Score information over purchases in Colombia

 The abstract now reads as:

Recently, front-of-package (FOP) food labeling systems have captured the attention of researchers and policy makers. Several Latin American governments are currently considering employing different FOP labeling systems. However, there is much need for more research-based evidence in these countries. In this paper, we study whether the immediate food-purchasing decision and the nutritional quality of the purchase are influenced by randomly informing some customers and not others about an FOP label known as Nutri-Score. We also separate the information effect from the effect of being aware of the system. We combined a randomized field intervention in a university cafeteria in Bogotá, Colombia with data from an after-purchase survey and receipts. We found that randomly providing information on Nutri-Score increased total expenditure by $0.18. Additional spending on healthier items was 21% or $0.26 higher, with no change for less healthy items. Expenditure estimates were higher among customers aware of the system’s existence. Treated customers were also 10% more likely to buy a healthier item than control customers were, and the protein content of their purchases was greater. Information on the Nutri-Score system increased the store’s sales. This potential financial incentive may ease implementation of Nutri-Score.

 In the Results Section, table titles have been changed to emphasize providing Nutri-Score Label information, for example:

Table 3. ITT and LATE effects of providing Nutri-Score Label information on total expenditure and expenditure by color, using different estimates.

Also, the results have been interpreted in this spirit. For example:

 Table 2 Column (1) shows that randomly providing Nutri-Score Label information increased total expenditure by US$ 0.18 (significant at the 5% level).

 When we refer to LATE results we do so in this fashion:

Row B of Table 3 shows LATE estimates. Column (1) shows that being aware of the system increased expenditure by US $0.29, much higher than the ITT estimate (the effect of the information) of US $0.18.

Row B [of Table 5] shows LATE on the non-complier subpopulation of successfully “treated” subjects. The estimates on Columns (1) and (6) shows that being aware of the Nutri-Score system made consumers…

 Point 1.2: You also need to focus on the previous literature (in a brief and concise manner) that is relevant to your study (treatment = providing information, labeling & cafeteria setting!). Only then, you can be more clear about what you are doing and your contribution.

 Response 1.2: We have reordered and added literature on providing information, labeling, and the cafeteria setting in the literature review section. The beginning now reads as follows:

 Nutrition Labeling and Information Provision

Nutrition labeling has taken on many forms in attempt to change people’s behavior: point-of-purchase menus[1] and back-of-package[2] (BOP) and front-of-package (FOP) nutrition labeling[3].

Additionally, a review by Seymour et al. (2004) on point-of-purchase environmental interventions observed that most interventions used information strategies. Overall, included studies were observed to increase sales of targeted food or change diets in a positive way. The reviewers also found that these interventions were most effective at worksites and universities, where alternative options were limited.

Focusing on the context of cafeterias, similar to our context, and information provision, Harnack & French (2008) reviewed studies that involved calorie labeling of menus in restaurants and cafeterias. They found that although there was some evidence of reducing calories purchased, the evidence had been inconsistent and weak, and the reviewers called for more rigorous studies in this area. An update to this review, focusing on quick-service restaurants, concluded that calorie labeling on menus did not decrease calorie ordering and consumption in included studies (Swartz, Braxton, & Viera, 2011). To note, Wisdom, Downs, & Loewenstein (2010), using a more experimental approach where participants were given the food they chose without knowing it was still part of the study, found that calorie information significantly reduced caloric intake.

More recently, studies of information provision of calories on menus at full-service sit-down restaurants has had promising results. Ellison, Lusk, & Davis (2014) found that providing calorie information combined with a traffic-light interpretation was able to reduce calories ordered at their restaurant on a university campus. Furthermore, Cawley, Susskind, & Willage (2018), in their working paper, conducted a randomized controlled field experiment using two full-service sit-down restaurants on a university campus. The authors found that providing calorie count information reduced calories ordered.

Our paper focuses on implementing an FOP nutrition label and randomly providing information about it in a university shop that sells food products. As mentioned above, the most relevant and most-used classification methodologies for FOP labels are detailed in Annex 1. From these labeling systems, we chose the Nutri-Score label for our intervention.

 Point 1.3: (By the way Line 73 “Two exceptions that use controls are” is not fully correct. Nikolova & Inman also have a control group comparison as robustness check!).

 Response 1.3: Thank you for pointing this out. We have modified the first sentence to include Nikolova & Inman and added a sentence as follows:

 Three exceptions that use controls are Nikolova & Inman (2015); Rahkovsky, Lin, Lin, & Lee (2013); and Surkan et al. (2016). Nikolova & Inman (2015) used a control group comparison as a robustness check in their study using scanner data from a large grocery chain before and after NuVal label implementation.

 Point 2.1: I am not convinced by the way you impute missing values for GPA. First, your statement in Footnote 8 is incorrect, namely “there is no valid approach to treat missing values, but dropping the observations with missing values”. The simplest way is to impute missing values by the average value of the non-missing values* (i.e. mean GPA; in the case there are not systematics = MCAR)! I think using the mean should be fine, as it is just a control variable. How do the results look then? If the results are unaffected by the value of your imputation, then use the mean and exclude that additional dummy variable from your control variables! From what I see in the Reviewer Response (p. 11), your results excluding the observations with missing GPA look substantially different than the ones reported in the paper (significant effects for all groups)! That gives the impression that your imputation (plus that additional dummy variable as control) have a too large and biasing impact on your results.

 Response 2.1: In the following results, you can see what happens when we impute the missing values in GPA by the average value of the non-missing values in GPA and omit the dummy variable to control for those cases:

Table 2. Unconditional effect of providing Nutri-Score Label information on total expenditure and expenditure by color. Intention to treat (ITT) estimates.  

Total

Green

Light
  Green

Orange

Pink

Red

Non-Labeled

(1)

(2)

(3)

(4)

(5)

(6)

(7)

Estimated Effect

0.171**

0.204***

0.060

-0.037

0.006

0.037

-0.099**

[0.077]

[0.074]

[0.034]

[0.034]

[0.036]

[0.045]

[0.045]

N

484

484

484

484

484

484

484

Notes: Each column shows the results of separate regressions. The dependent variable in Column (1) is the amount of money spent by a consumer on his/her purchase, in Column (2) is the amount of money spent in green items only by a consumer in his/her purchase, etc. If a consumer did not buy any green items, then the amount of money spent is taken as zero. All specifications include the following full set of pre-determined controls: individual controls (GPA, age, freshman, estrato, BMI, breakfast, nutritional facts use, smoker, drinker and sex), day of week and week number. Robust standard errors in parenthesis. ** p<0.05 *** p<0.01< p="">

Table 3. ITT and LATE effects of providing Nutri-Score Label information on total expenditure and expenditure by color, using different estimates.

Total

Green

Light
  Green

Orange

Pink

Red

Non-Labeled

(1)

(2)

(3)

(4)

(5)

(6)

(7)

A. Intention to treat (ITT)

0.171**

0.204***

0.06

-0.037

0.006

0.037

-0.099**

[0.077]

[0.074]

[0.034]

[0.034]

[0.036]

[0.045]

[0.045]

N

484

484

484

484

484

484

484

B. Local average treatment effect   (LATE)

0.277**

0.331***

0.098

-0.060

0.010

0.060

-0.161**

[0.121]

[0.117]

[0.054]

[0.053]

[0.057]

[0.071]

[0.070]

N

484

484

484

484

484

484

484

Notes: This table follows the same specification as Table 2. Robust standard errors in parenthesis. ** p<0.05 *** p<0.01< p="">

Table 4. Effect of providing Nutri-Score Label information on expenditure by color, conditional on buying at least one item of a given color.

Green

Light
  Green

Orange

Pink

Red

Non-Labeled

(1)

(2)

(3)

(4)

(5)

(6)

Estimated Effect

0.341**

0.673

-1.703

-0.007

0.046

-0.047

[0.143]

[0.320]

[0.923]

[0.115]

[0.047]

[0.050]

N

137

33

32

83

223

179

Notes: The dependent variable in this table is the amount of money spent by each person (or on each purchase) on items of the indicated color in each column, conditional on buying at least one item of the same color. This table uses the same covariates as in Table 2. Robust standard errors in parenthesis. ** p<0.05 *** p<0.01< p="">

Table 5. ITT and LATE effects of providing Nutri-Score Label information on the probability of purchasing at least one item of a particular color using a linear probability model (LPM).

Green

Light
  Green

Orange

Pink

Red

Non-Labeled

(1)

(2)

(3)

(4)

(5)

(6)

A. Intention to Treat (ITT)

0.099**

0.026

-0.011

-0.002

0.016

-0.098**

[0.042]

[0.023]

[0.024]

[0.035]

[0.045]

[0.045]

N

484

484

484

484

484

484

B. Local average treatment effect   (LATE)

0.161**

0.041

-0.018

-0.003

0.026

-0.159**

[0.066]

[0.036]

[0.038]

[0.055]

[0.071]

[0.071]

N

484

484

484

484

484

484

Notes: The dependent variable in this table is a dummy that takes the value of one if the person bought at least one item of the color indicated in each column, and zero otherwise. All other specifications are the same as in Table 2. Robust standard errors in parenthesis. ** p<0.05 *** p<0.01< p="">

Table 6. ITT and LATE effects of providing Nutri-Score Label information over nutritional content of bought items in a standardized serving size (100 g for solid or 100 ml for liquid), by nutrient.

Proteins
  (g)

Calories
  (kcal)

Sugars
  (g)

Sodium
  (mg)

Saturated
  Fats (g)

Fibers
  (g)

(1)

(2)

(3)

(4)

(5)

(6)

A. Intention to Treat (ITT)

2.812***

66.672

-2.677

38.074

0.031

0.654

[0.856]

[111.434]

[2.505]

[25.123]

[1.218]

[0.578]

N

393

393

393

393

393

393

C. Local average treatment effect   (LATE)

4.421***

104.826

-4.210

59.863

0.049

1.028

[1.319]

[169.372]

[3.834]

[38.181]

[1.854]

[0.882]

N

393

393

393

393

393

393

 Notes: Robust standard errors in parenthesis. ** p<0.05 *** p<0.01< span="">

 As you can see, the results are quite similar to those in the paper. On the other hand, if we impute missing values in GPA by the average value of the non-missing values in GPA and include the dummy variable to control for those cases, then the results look as those in the paper. Again, it does not matter what values are imputed, as long as we control for those cases using the dummy. The dummy may be increasing the standard errors, but we prefer our approach.

 We are changing footnote 8 as follows:

 We created a dummy variable that takes the value of 1 when the GPA value is missing, 0 otherwise. When the GPA is missing we input a value of 99. The full set of covariates in the regressions include the modified GPA and the dummy variable. The results are the same, regardless of the inputted value, as long as we add the dummy variable as a covariate. We also tried the following approaches: Dropping the observations with missing values in GPA (n=75), which affects our standard errors and make the p-values smaller, but the coefficients are similar to those in Table 2. Imputing missing values in GPA by using the average value of the non-missing values in GPA and excluding the additional dummy variable from our covariates, as suggested by an anonymous referee, which affects our coefficients slightly but statistical significance remains the same. These last results are presented in the Online Appendix.

Regarding this sentence:

 From what I see in the Reviewer Response (p. 11), your results excluding the observations with missing GPA look substantially different than the ones reported in the paper (significant effects for all groups)!

 Please notice that * p<0.1 in the table you mention in p.11, which is not significant according to the other Reviewer. Hence, there are no significant effects for all groups. We did not erase the “*”, we apologize for this mistake. The table in p. 11 should read as follows:

 Unconditional effect of providing Nutri-Score Label information on total expenditure and expenditure by color. Intention to treat (ITT) estimates.

Total

Green

Light Green

Orange

Pink

Red

Non-Labeled

(1)

(2)

(3)

(4)

(5)

(6)

(7)

Estimated
  Efffect

0.125

0.159**

0.064

-0.063

-0.015

0.081

-0.101**

[0.082]

[0.080]

[0.039]

[0.036]

[0.039]

[0.049]

[0.049]

N

409

409

409

409

409

409

409

 Notes: In this Table, missing values in GPA are left as missing. Robust standard errors in parenthesis. ** p<0.05 *** p<0.01< p="">

 *In addition, there is regression-based imputation (i.e., predict missing GPA values from a linear regression with GPA as dependent variable and strong explanatory variables from where GPA values are not missing) and the EM-algorithm proposed by Dempster, Laird, & Rubin (1977) [Maximum likelihood from incomplete data via the EM algorithm. Journal of the Royal Statistical Society]. The latter two approaches are just for your information!

 Response: Thank you for letting us know this, we deeply appreciate your experience on this topic.

 Point 3.1: The main result is increased spending in the green category and decreased spending in the unlabeled category. Perhaps the first result comes from the content of the news article and would look different if your treatment would be supplemented with different information. For example, what about an article that emphasizes health problems due to high sugar consumptions? Would that perhaps lead to a decrease in the red category? Please interpret your results in this light and also discuss this issue (also in your limitations).

 Response 3.1: This is an excellent point and we agree with the Reviewer. However, this was the reason why we gave the same news article to both the treatment and control groups. If there were any effect of the news article over consumption decisions, it would affect both groups equally, meaning any differences we find between groups would be in addition to the effect of the news article.

 To make this point clearer, we modified the following paragraph at the beginning of subsection 2.3 Intervention Materials (notice that we deleted footnote 6):

 We had two types of laminated information placards. Figure 1 (a) shows our control placard, and Figure 1 (b) shows the treatment placard. The control and treatment placards contained a news article related to processed food consumption in Latin American countries[4]. We chose a news article related to nutrition in order to make control and treatment placards as comparable as possible. The reason why we gave the same news article to both treatment and control groups is controlling for the content of the news article itself. If there were any effect of the news article over consumption decisions, then it would affect both groups equally, meaning any differences we find between groups would be in addition to the effect of the news article. The difference between the treatment and control placard was that the information on the Nutri-Score labeling system existence was present only in the treatment placard.

 We also changed the paragraph at the beginning of Section 4. Discussion:

 Our overall results indicate that those in the treated group bought similar items to the control group and at least one additional green item, which resulted in those in the treated group buying items containing more protein. Since both treatment and control groups were given the same news article, we do not believe these results are due to the content of the news article and these results would look differently if our treatment were instead supplemented with different information. Below we organize our discussion by our specific objectives.

 Point 4.1: Your results suggest more calorie intake by the treatment! Be more clear that it is not a desirable outcome and that purchases of unhealthful items did not decrease, and what this means in particular to public policy when it comes to considering your concrete treatment (information provision!).

 Response 4.1: We modified the last paragraph of Section 3. Results to the following:

Table 6 presents regression results for ITT and LATE estimates, using the full set of controls, and a standardized serving size of 100 grams for solid food or 100 milliliters for beverages. Row A shows that randomly providing the Nutri-Score Label information increased the protein content of purchases by 2.73 grams. Row B shows that being aware of the system increased protein purchase by 4.31 grams. There is no statistically significant change in the purchase of other nutritional content. However, the point estimate in Column (2) shows that calorie content of purchases increased, which was not a desirable outcome, in addition to purchases of unhealthful items not decreasing. Therefore, providing information to customers about the meaning of the Nutri-Score Label does not necessarily decrease the calorie content of their purchases.

 In the discussion, we added the following sentence in Section 4.3:

However, there were no differences between treatment and control in probability of buying at least one product labeled with the other less healthy colors.

And added the following paragraph in Section 4.4:

However, treated customers also purchased more calorie contnet compared to control customers. This result was not desirable and is contrary to the results of Cawley et al. (2018); Ellison et al. (2014); and Wisdom et al. (2010), who all found that providing calorie count information (for Ellison et al. (2014) only in conjunction with a traffic light interpretation) decreased calories purchased. This, coupled with the fact that there were no differences between groups in the likelihood of purchasing less healthy items, gives us pause. There is a need for more research before implementing any policy with the Nutri-Score label along with information provision about the label, in our context.

 Point 5. Please consider to employ a professional English-proofreader, as the manuscript still has a lot of grammatical errors.

 Response 5: A professional English proofreader has now edited the manuscript.

[1] For examples, see: (Harnack & French, 2008; Kiszko, Martinez, Abrams, & Elbel, 2014; Seymour et al., 2004; Sinclair, Cooper, & Mansfield, 2014; Swartz et al., 2011; Vyth et al., 2011)

[2] For examples, see: (Berning, Chouinard, & McCluskey, 2008; Cioffi, Levitsky, Pacanowski, & Bertz, 2015; Gracia, Loureiro, & Nayga, 2007; Kim, Nayga, & Capps, 2000; Loureiro, Gracia, & Nayga, 2006; Variyam, 2008; Variyam & Cawley, 2006)

[3] For examples, see: (Andrews et al., 2014; Aschemann-witzel et al., 2013; Balcombe et al., 2010; Chiuve, Sampson, & Willett, 2011; Ducrot et al., 2016; Feunekes, Gortemaker, Willems, Lion, & van den Kommer, 2008; Julia et al., 2016; Katz et al., 2010; Levin, 1996; Levy et al., 2012; Rahkovsky et al., 2013; Sacks et al., 2009; Schucker et al., 1992; Sonnenberg et al., 2013; Steenhuis et al., 2010; Sutherland et al., 2010; Thorndike et al., 2014, 2012; Waterlander, Steenhuis, De Boer, Schuit, & Seidell, 2013)

[4] http://www.bancomundial.org/es/news/feature/2018/01/10/comer-fuera-para-muchos-significa-ir-a-un-puesto-de-comida-callejera?cid=ECR_E_NewsletterWeekly_ES_EXT&deliveryName=DM1765

Reviewer 2 Report

The paper is now much improved. However the following edits need to be made.

Line 42, change “incidence” to “prevalence”.

Line 456, change “system” to “system’s”.

Line 462, change “Each column in this table show” to “Each column shows”.

Line 553, change “in” to “on”, twice.

Lines 554-555, reword as follows:  “The main takeaway is that among all subjects who were assigned to treatment, expenditure increased by $0.18, but among those assigned to treatment and who were also aware of the existence of the system, expenditure increased by $0.29.”

Line 557, what is “intensive margins”?

Line 639, change “proteins” to “protein”. Same in Figure 6 and line 755.

Figure 6 and Table 6, the usual abbreviation for grams is g, not gr. In other places the authors use the abbreviation g. Be consistent.

Line 658, change indicated to indicate.

Line 660, delete “content”.

Line 755, change “saturated fats” to “saturated fat”.

Change “sugars” to “sugar” in several places.

Line 757, change “more protein concentration” to “a higher concentration of protein”

Line 764, probably change “normal” to “unhealthy”

Line 778, start new sentence at “These include…”

Line 812, change “like” to “as occurred”

Line 814, change “thought” to “believe”

 Author Response

Response to Reviewer 2 Comments

 The paper is now much improved. However the following edits need to be made.

Thanks you for your edits. We have incorporated them.

 Point 1. Line 42, change “incidence” to “prevalence”.

 Response 1: This has been done and the sentence now reads:

 The Colombian case is no exception and the incidence of these conditions have significantly grown.

 Point 2: Line 456, change “system” to “system’s”.

Response 2: This has been done and the sentence now reads:

 If a person answers “I don’t know what it is” then we classify that person as not aware of the system’s existence.

 Point 3: Line 462, change “Each column in this table show” to “Each column shows”.

 Response 3: This has been done and the sentence now reads:

 Each column shows the results of separate regressions.

 Point 4. Line 553, change “in” to “on”, twice.

 Response 4: This has been done and the sentence now reads:

 The LATE increase is due to a US $0.33 increased expenditure on green items, and a decreased expenditure on non-labeled items of US $0.15.

 Point 5. Lines 554-555, reword as follows:  “The main takeaway is that among all subjects who were assigned to treatment, expenditure increased by $0.18, but among those assigned to treatment and who were also aware of the existence of the system, expenditure increased by $0.29.”

 Response 5: This has been reworded as you suggested.

 Point 6. Line 557, what is “intensive margins”?

 Response 6: Thanks for pointing this out. We have changed the writing in several Lines to make this clearer. Lines 367-372 now reads as follows:

 The main outcome of interest is nominal expenditure (overall, and on items of a given color classification). We also analyze the extensive and intensive margin of nominal expenditure on items of a given color classification; i.e. whether the customer ordered at least one item of a given color and how much money the customer spent in that color conditional on buying at least one. The last outcome is the nutritional content per purchase (calories, saturated fat, fiber, protein, sodium and sugar).

 Also, Lines 557-563 now reads as follows:

 We now move to analyze the extensive and intensive margins by classification color; the extensive margin is the probability of buying any item of a given color (green, light green, etc.) and the intensive margin is the amount of money spent on items in that color category conditional on buying any. Remember that a person can buy one item (or many items) of a given color or none at all. We start with the intensive margin in Table 4. Column (1) shows that, conditional on buying at least one green item, the intervention increased expenditure on green items by US $ 0.34. There is no statistical evidence that expenditure decreased on the less healthy items (red and pink), or those with missing labels. Table A.2 shows robustness results for the effect of the intervention on total money spent and on the natural logarithm of total money spent, with and without covariates.

 Lines 610-611 now reads as follows:

 We now study what happened with the demand, or the likelihood of purchasing an item of a given color, i.e. the extensive margin.

 Lines 616-617 now reads as follows:

 Table 5 shows different estimates of the extensive margin, the likelihood of purchasing at least one item, by color, using the full set of controls, and a linear probability model.

 Point 7: Line 639, change “proteins” to “protein”. Same in Figure 6 and line 755.

 Response 7: This has been done and the first sentence now reads:

 Treated group purchased on average more protein than the control group.

 Figure 6 now reads, Protein.

 And the last sentence now reads:

 Quantify if the nutritional quality of items purchased between treated and control participants is different in terms of calories, protein, saturated fat, fibers, sodium and sugar.  

 We also did a search and changed all references of proteins to protein (there was one additional instance).

 Point 8. Figure 6 and Table 6, the usual abbreviation for grams is g, not gr. In other places the authors use the abbreviation g. Be consistent.

 Response 8: You’re right, Figure 6 caption now reads:

Nutritional content of purchases, by nutrient and randomization group, using a standardized serving size of 100 g or ml.

And Table 6 title now reads:

ITT and LATE effects of the Nutri-Score Label over nutritional content of bought items in a standardized serving size (100 g for solid or 100 ml for liquid), by nutrient.

 Point 9: Line 658, change indicated to indicate.

 Response 9: This has been done and the sentence now reads:

 Our overall results indicate that those in the treated group bought similar items to the control group and at least one additional green item, which resulted in those in the treated group buying items containing more protein.

 Point 10: Line 660, delete “content”.

 Response 10: This has been done and the sentence reflecting this change is pasted above, in Response 9.

 Point 11: Line 755, change “saturated fats” to “saturated fat”.

 Response 11: This has been done and the sentence now reads:

 Quantify if the nutritional quality of items purchased between treated and control participants is different in terms of calories, protein, saturated fat, fibers, sodium and sugar.  

 We also did a search and changed all instances of fats in the text to fat.

 Point 12: Change “sugars” to “sugar” in several places.

 Response 12: This has been done, see the above instance in Response 11. We also did a search and changes all instances of sugars in the text to sugar.

 Point 13: Line 757, change “more protein concentration” to “a higher concentration of protein”

 Response 13: This has been done and the sentence now reads:

 This was due to the fact that the green-labeled items available overall contained a higher concentration of protein than items labeled with other colors.

 Point 14: Line 764, probably change “normal” to “unhealthy”

 Response 14: “less healthy” is more accurate, so we’ve incorporated it as follows:

 Although we did not look at actual consumption, treated customers may have then eaten less of the less healthy item they bought…

 Point 15: Line 778, start new sentence at “These include…”

 Response 15: This has been done.

 Point 16: Line 812, change “like” to “as occurred”

 Response 16: This has been done, the sentence now reads:

 It may be the case that customers bought extra healthier items, but then threw them away as occurred in a short-term school convenience-line intervention (Hanks et al., 2012)

 Point 17: Line 814, change “thought” to “believe”

 Response 17: This has been done, the sentence now reads:

 The authors believed, however, that additional exposure to the healthier food on students’ trays would eventually result in increased consumption of the healthier food as well.